# Evolution of Proliferative Model Protocells Highly Responsive to the Environment

**DOI:** 10.3390/life12101635

**Published:** 2022-10-19

**Authors:** Muneyuki Matsuo, Taro Toyota, Kentaro Suzuki, Tadashi Sugawara

**Affiliations:** 1Department of Chemistry, Graduate School of Integrated Sciences for Life, Hiroshima University, Kagamiyama, Higashi-Hiroshima 739-8526, Japan; 2Department of Basic Science, Graduate School of Arts and Sciences, The University of Tokyo, 3-8-1 Komaba, Meguro-ku, Tokyo 153-8902, Japan; 3Department of Chemistry, Faculty of Science, Kanagawa University, Tsuchiya, Hiratsuka 259-1293, Japan

**Keywords:** constructive approach, membrane, compartment, giant vesicle, proliferation, information flow, phenotype, DNA, lipo-deoxyribozyme, predominate species, supramolecular protocell machine

## Abstract

In this review, we discuss various methods of reproducing life dynamics using a constructive approach. An increase in the structural complexity of a model protocell is accompanied by an increase in the stage of reproduction of a compartment (giant vesicle; GV) from simple reproduction to linked reproduction with the replication of information molecules (DNA), and eventually to recursive proliferation of a model protocell. An encounter between a plural protic catalyst (**C**) and DNA within a GV membrane containing a plural cationic lipid (**V**) spontaneously forms a supramolecular catalyst (**C@DNA**) that catalyzes the production of cationic membrane lipid **V**. The local formation of **V** causes budding deformation of the GV and equivolume divisions. The length of the DNA strand influences the frequency of proliferation, associated with the emergence of a primitive information flow that induces phenotypic plasticity in response to environmental conditions. A predominant protocell appears from the competitive proliferation of protocells containing DNA with different strand lengths, leading to an evolvable model protocell. Recently, peptides of amino acid thioesters have been used to construct peptide droplets through liquid–liquid phase separation. These droplets grew, owing to the supply of nutrients, and were divided repeatedly under a physical stimulus. This proposed chemical system demonstrates a new perspective of the origins of membraneless protocells, i.e., the “droplet world” hypothesis. Proliferative model protocells can be regarded as autonomous supramolecular machines. This concept of this review may open new horizons of “evolution” for intelligent supramolecular machines and robotics.

## 1. Introduction

To understand the complex dynamics of life, the “constructive approach”, characterized by the construction of an appropriate model system using relatively simple substances with well-defined properties, is a promising methodology. In particular, Luisi profoundly discussed the roles of model systems composed of well-defined molecules in terms of the emergence of life [1,2]. Stano proposed an extension of the constructive approach using synthetic cells for networks and complex structures [3]. If a synthesized model system, such as a primitive model cell, successfully reproduces a repeatable proliferation in which self-reproduction of a giant vesicle (GV) and replication of DNA are linked, it can be said that life dynamics are elucidated. The essential life dynamics to be focused on include compartmentalization, autonomous movement, proliferation, primitive flow of information, phenotypic plasticity corresponding to the environment, natural selection, and evolution. Because a living cell possesses these properties, it can be regarded as the ultimate supramolecular machine [4]. In this review, we discuss various methods of reproducing the aforementioned life dynamics using a constructive approach.

## 2. Self-Reproductive Model Protocell

### 2.1. Definition of GV Self-Reproduction and Some Examples

As discussed in the introduction, to understand the self-reproduction dynamics of model protocells, it is promising to construct a supramolecular system that self-reproduces by incorporating a nutrient. Here, we discuss why GVs made of amphiphiles are appropriate as compartments of a model protocell.

GVs are formed spontaneously by the self-assembly of amphiphiles in water, and they can encapsulate ions or hydrophilic molecules in their interior water pool during the construction of a compartment. If a hydrophobic molecule, such as a hydrophobic nutrient, is caught in micelles that disperse in the outer water phase, it can be transferred into a vesicular membrane. Hence, this supramolecular system is expected to exhibit higher-order dynamics and approach a primitive model protocell (Figure 1) [1,2,3,5]. In studies focused on the construction of self-reproducible molecular systems, chemical methods are advantageous because of the high degrees of freedom in the molecular structures that can be controlled experimentally [6]. Luisi proposed the following definition for a self-reproductive artificial cell [1]: consider a vesicle composed of membrane molecule **S**. If a vesicle takes in membrane precursor **A** and converts it into **S** in the presence of catalyst **C** within the vesicle, the enlarged GV becomes destabilized and is divided into two equivolume GVs. Such dynamics can be regarded as “self-reproduction”.

In research on primitive model cells, a GV with a diameter larger than 1 μm is often used because it is large enough to be observed by optical microscopy. In a pioneering work on self-reproductive GVs, Walde et al. found that the hydrolysis of oleic anhydride proceeds autocatalytically in the presence of oleic acid in basic water [6]. They also found that the number of preexisting GVs increased under this condition. However, it is not clear as to whether all oleic anhydrides are hydrolyzed within the GV membrane because the hydrolysis of oleic anhydride also proceeds in basic water.

Stano and Luisi also discussed the size features of newly formed vesicles made of oleic acid in the presence of pre-existing GVs. When fatty acid micelles are incorporated into preexisting vesicles, these vesicles become enlarged, unstable, and finally split. To some extent, the particle size of the split vesicle reflected that of the original vesicle. This phenomenon is called the “matrix effect” [7].

Zhu and Szostak described a simple and efficient pathway for growth and division of a model protocell membrane. When fatty acid micelles are supplied to a dispersion of large multilamellar fatty acid vesicles, these vesicles transform into long, thread-like vesicles. The application of modest shear forces is sufficient to divide the thread-like vesicles into multiple spherical daughter vesicles. The encapsulated RNA molecules were also found to be distributed in the daughter vesicles [8]. Despite these excellent studies, a self-reproductive GV satisfying Luisi’s definition has not yet been reported.

### 2.2. Self-Reproduction of GVs Containing a Catalyst Induced by Addition of a Nutrient

The authors reported two clear examples of GV-based model protocells that satisfied Luisi’s definition. An external raw material for a membrane molecule was taken from the outside and converted to a membrane molecule inside the GV (birthing-type self-reproduction) [9] or in the GV membrane (budding-type self-reproduction) [10] under the presence of a catalyst. When the number of membrane molecules was increased, a morphological change was induced, and vesicles with almost the same particle size were reproduced despite the absence of a template molecule. Membrane dynamics play a crucial role in GV deformation and division in both types of self-reproduction.

In birthing-type self-reproduction, membrane lipids (**V_bi_**) are synthesized by dehydrocondensation [**H** (–NH_2_) + **T** (–CHO) → **V_bi_** (–N=CH–)] between a hydrophilic head unit (**H**) taken from the outside on the surface of an oil droplet composed of hydrophobic tail units (**T**) encapsulated in a GV. The resulting membrane molecule self-assembles to produce daughter vesicles in the mother GV. The daughter GV is pushed outward in conjunction with the membrane contraction movement of the mother vesicle, thereby completing GV self-reproduction [9]. In budding-type self-reproduction, the membrane precursor, bolaamphiphile (**V***), is hydrolyzed to produce membrane lipids (**V**) and electrophiles (**E**) [**V*** (–N=CH–) → **V** (–CHO) + **E** (–NH_2_)] within the membranes of giant multilamellar vesicles (GMVs) containing the catalyst (**C**) (Figure 2a). The GMV (giant multi-lamella vesicle) grows and is squeezed in the middle before dividing into two daughter GMVs, both of which are almost identical in volume to the original, mother, GMV [10].

Essentially, the purpose of using a bolaamphiphile, which is an amphiphile bearing hydrophilic units at both ends of the hydrophobic chain, as membrane precursor **V***, is as follows. Because **V*** dissolves unimolecularly in water and does not form micelles or vesicles, it is easily incorporated into pre-existing GMVs. Hence, a series of GMV dynamics (enlargements, budding deformations, and equivolume divisions) are completed within approximately 20 min after the addition of **V*** (Figure 2b) [10]. Umeda proposed an elastic model for the dividing of GMVs [11].

**Figure 2 life-12-01635-f002:**
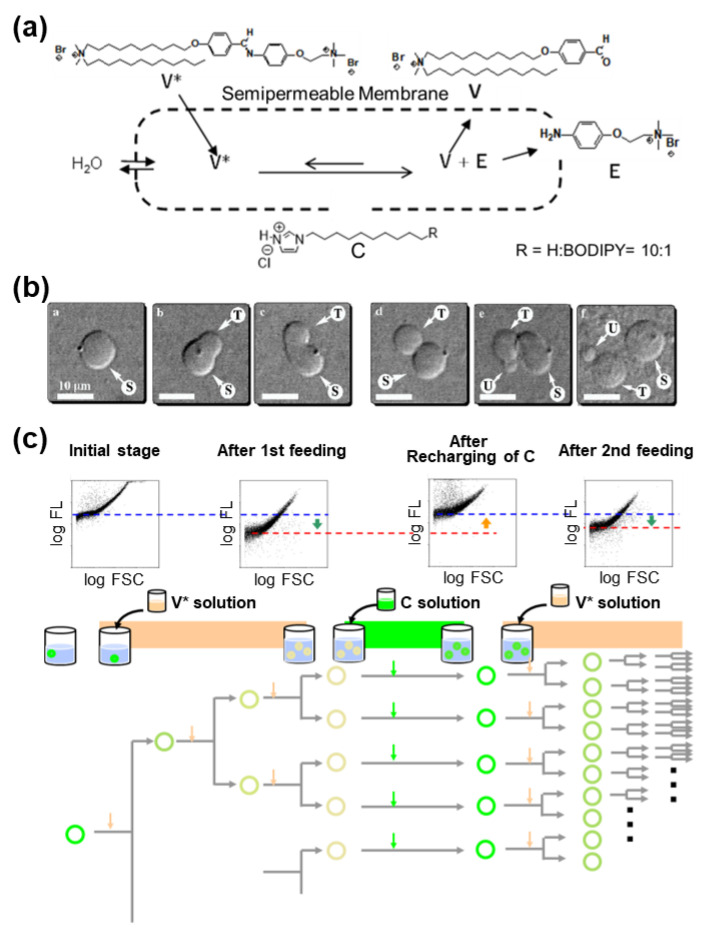
(**a**) Molecular structures and reaction scheme of self-reproduction of a GMV. (**b**) Microscopy observation of self-reproduction of a GMV. Adapted with permission from Ref. [10]. Copyright 2004 American Chemical Society. (**c**) Population analysis of GV self-reproduction base on the flow-cytometry measurements. At 2 h after the addition of the precursor, a catalyst solution was added. The amount of **C** in the vesicles was recovered, and the GV production was restarted the by addition of the precursor. Adapted with permission from Ref. [12]. Copyright 2006 American Chemical Society.

The population analysis of self-reproductive GMVs by flow cytometry is shown in Figure 2c. The amount of protic catalyst **C** was measured by the fluorescence intensity of a fluorescent probe in the membrane, which decreased by approximately one-tenth after three or four divisions. However, when **C** was added to the outer aqueous phase, the membrane was refilled with **C** (Figure 2c), and the reproduction of the GMVs started again with the addition of **V***. Compared with the original GMVs, the number of amplified GMVs increased by approximately 100 times [12].

Although the proliferation of a model cell has been demonstrated, the method of cell division may be too simple compared with that of a living cell. That said, Errington et al. reported thought-provoking results relating to exactly this problem. In some modern bacteria, when the bacteria infect the host, the cell wall breaks down to form an L-shape. The cells do not require the protein machinery called FtsZ, which is necessary for cell division, instead only membrane lipid synthase is activated, causing the membrane to elongate into a tube-like shape and divide itself owing to the strain caused by the imbalance between the surface area and the volume. The “squeeze to divide-type” artificial cell described above is an appropriate model for this type of cell division [13,14].

### 2.3. Sustainable Self-Reproducing GV Consisting of a Synthetic Phospholipid

If the vesicle membrane itself acts as a catalyst, there is no need to add a catalyst or worry about catalyst depletion due to splitting, resulting in the high self-sustainability of proliferative GVs. The phosphate anion in the buffer catalyzes the hydrolysis of the phosphate ester. The self-reproduction of phospholipid GVs was examined by adding a synthetic phospholipid precursor to a phosphate buffer dispersion (Figure 3a). As the phosphoric membrane surface of the pre-existing GV functioned as a catalytic field in phosphate buffer (pH 7.7–5.7), membrane lipids were generated in an autocatalytic manner [15].

The membrane area increased with the addition of the synthetic phospholipid precursor, **V^−^*****_ole_**; however, the change in GV volume was small. The surplus membrane causes the GV to squeeze and eventually split (Figure 3a) [15]. This outcome was achieved by modifying a previously reported phospholipid [16] by replacing the dodecyl group with an unsaturated alkyl (oleyl) group. Enzymatic DNA replication (polymerase chain reaction, PCR) was achieved in this sustainably self-reproductive GV because the GVs exhibited thin lamellarity and contained an inner aqueous solution [16]. Hence, this anionic mono-component GV made of synthetic phospholipid **V^−^****_ole_** can be regarded as a “self-sustainable proliferating machine” (Figure 3b,c).

Devaraj et al. described a unique phospholipid production system in a vesicular membrane accompanied by the formation of a catalyst [17,18]. The GV membrane contains amphiphilic tridentate ligands that are complex with Cu (I) ions to form a catalyst. In contrast, a two-legged membrane lipid is formed through the Huisgen cycloaddition of lauryl azide with a single-chain phospholipid bearing an ethynyl unit in the presence of the Cu (I) ion catalyst. Therefore, it is possible to continuously produce phospholipids by the addition of lipid precursors and Cu (I) salts. However, in their reports, because the original phospholipid of GVs is natural and different from the products, the self-reproductive ability of this phospholipid production system has not been achieved.

## 3. Mechanisms of Morphological Changes of GV-Based Protocells

### 3.1. Interpretation of Morphological Changes of GVs by the Area Difference Elasticity (ADE) Model

In Section 2.2, GV self-production is discussed. A curious issue is that the size of the generated GV (daughter GV) is almost the same as the original GV (mother GV), even without a template, regardless of birthing- or budding-type production. There is a rule that the curvature of the GV membrane is a function of a ratio of the cross-sectional area (*a*) of the lower part of the hydrophilic head of the amphiphile and the volume (*V*) of the hydrophobic tail. The ratio *a*/*V* is called a packing parameter, *p*, which roughly determines the shape of molecular assemblies, for example, micelles and vesicles [19].

The area difference elasticity (ADE) theory precisely rationalizes the shape of molecular assemblies, as discussed below. GV morphology changes in response to physical stimuli, such as changes in temperature or osmotic pressure; the ADE model seeks to explain the various shapes and morphologies of GVs [20,21,22]. In the ADE model, the difference in the number of molecules (or area) between the inner and outer leaflets of the membranes (*N*_out_ − *N*_in_ or *A*_out_ − *A_i_*_n_ = ∆*a*, where *N* is the number of molecules and *A* is the area) is a key variable in relation to their elastic stability.

Imai and colleagues presented a unilamellar vesicular model containing an “inverse-cone” structured amphiphilic molecule with a large packing parameter, *p* [19,23]. This vesicle transformed into a limited form upon heating (Figure 4a). This was because the addition of an inverse-cone amphiphile stabilizes the limited form **E**, as shown in Figure 4b [24], where the two spheres are connected by a very thin neck. The entire process of this type of GV was rationalized using ADE analysis; the GV eventually split into two vesicles during the reheating stage [24]. However, morphological changes in self-reproductive GVs induced by the added membrane precursor are difficult to analyze because of the participation of additional molecules. Hence, this morphological change has not yet been theoretically understood.

### 3.2. Time-Dependent Morphological Changes from Micelles to Vesicles

Toyota et al. reported continuous morphological changes in the self-assembly of amphiphiles (*N*-benzylidene aniline derivative), accompanied by hydrolysis of the amphiphile to produce hydrophilic and hydrophobic molecules in the membrane (Figure 5a) [25]. The morphological evolution of spherical micelles into nested GVs occurred spontaneously (Figure 5b). This continuous change can be explained by the gradual increase in the packing parameter *p* of the amphiphiles, as well as the aggregation of micelles and vesicles resulting from the ionic strength increase by the released head groups (Figure 5c). As hydrolysis proceeded, the generated hydrophobic product loosely interacted with the original amphiphile to form a dimer, which could be regarded as a pseudo-two-legged amphiphile. As a result, the *p* value of the weakly formed dimer increased, accompanied by hydrolysis, and the micelles were smoothly transformed into GVs because of the increase in the volume of the pseudo-two-legged amphiphile, **V**_practical_.

Later, Takakura et al. constructed an advanced model of the spontaneous transformation from micelles to vesicles, associated with the sequential conversion of amphiphiles [26]. The formation of true double-chained amphiphiles **V** and electrophile **E** proceeded smoothly through migration of the dodecylamino group from **N_1_** to **N_2_**. These results strongly support the aforementioned explanation of the model of Toyota et al. (Figure 6). In the equilibrium between **N_1_** + **N_2_** and **E** + **V**, the two-legged amphiphile **V** had a large *p* value, forming a GV as a molecular assembly. This morphological change was also explained by the ADE model.

### 3.3. Conversion of Tubular Phospholipid GVs to Spherical GVs by the Addition of a Precursor Lipid

Subsequent research focused on GVs composed of phospholipids with phosphocholine head groups because the major phospholipids in living cells are zwitterionic, such as 1-palmitoyl-2-oleoyl-*sn*-glycero-3-phosphocholine (POPC). Artificial zwitterionic phospholipids (**V_PC_**) and their precursors (**V*_PC_**) were prepared to elucidate the morphological stability of GV_PC_ (Figure 7a) [27]. **V*_PC_** is hydrolyzed into **V_PC_** in aid of the amphiphilic catalyst **C**. Before adding **V_PC_*******, the observed GV_PC_ (**V_PC_**:**C**: cholesterol = 7:1:2 molar ratio) had spherical and hollow structures with diameters greater than 10 μm. When a solution of **V_PC_*** with a concentration ≥0.5 mM was added to a dispersion of GV_PC_ on a heated stage at 45 °C, spherical GV_PC_ was transformed into tubular GV_PC_ (Figure 7b). However, tubular GV_PC_s were converted back to spherical GV_PC_s by heating to 60 °C via a one-way deformation. Macroscopically, the aforementioned results indicate that the spherical GV_PC_ is a thermodynamically controlled product, whereas the tubular GV_PC_ is formed as a kinetically controlled product (Figure 7c).

These experimental results can be rationalized by analyzing the elastic energies. At the initial stage, the elastic energies of the spherical and tubular GVs were almost the same, and the two types of GVs coexisted in equilibrium between the two forms. The bending energy of a spherical GV decreased in a nonlinear manner as the bending energy in two directions (*H*x, *H*y) of a vesicular membrane was inversely proportional to the square of the curvature radius-(1/*R*)^2^. In contrast, the bending energy of the tubular GV along the tubular direction (*H*t) remained constant during the elongation stage. While the elastic energy of the elongated tubular GV increased with its length, the elastic energy of the spherical structure with the same area decreased when accompanied by an increase in diameter.

The one-way transformation from a tubular GV to a spherical GV may be limited to the morphological changes in a phospholipid GV induced by the membrane precursor. This analysis is discussed further in Section 2.1 (long thread-like fatty acid vesicles) and Section 7.2 (phenotype plasticity).

## 4. Cooperative Self-Reproduction of GVs with Self-Replication of DNA

### 4.1. Encounter of DNA with Lipids and Enzymes

Researchers who study the origins of life using a constructive approach define a model protocell as the smallest unit in life with three essential elements: a partition that distinguishes the internal reaction system from the outside environment, enzymes that carry out metabolic reactions inside, and information molecules that are passed on individuality to the offspring (Figure 8) [28]. Furthermore, they claim that the emergence of heritable and evolvable self-reproduction is indispensable.

Sharov pointed out the interesting possibility that primordial life had no nucleic acids and that the heritable signs were represented by catalytically active self-reproducing coenzyme-like molecules, which were controlled by other polymers used as templates. This type of template-based synthesis eventually resulted in the emergence of RNA-like replication [29].

In this section, we discuss the proliferative GV-based model of protocell-encapsulated RNA or DNA. In a pioneering study, Luisi et al. constructed small vesicles (SVs, *ϕ* < 100 nm) composed of oleate/oleic-acid-containing RNA and Qβ-replicase. They succeeded in RNA replication and reproduction of small vesicles; however, cooperation between these processes has not been observed [30,31].

Hanczyc and Szostak reported that RNA adsorbed onto clay can be encapsulated within vesicles. Such vesicles grow by incorporating fatty acids supplied as micelles and can be divided by extruding them through small pores [32]. Mansy et al. prepared a semi-permeable small vesicle and used non-enzymatically synthesized short DNA (≈30 base pairs; bp) as activated nucleotides in this prebiotic cell [33]. Physical procedures were used to divide the vesicles. Although excellent examples have been presented, the intimate interplay between the self-reproduction of GVs and the self-replication of information molecules in GVs has been scarcely explored.

### 4.2. Proliferative GV-Based Model Protocell Encapsulating DNA

The encounter of DNA with the plural protic catalyst **C** of a lipophile in a giant vesicular membrane led to the formation of supramolecular catalysts (**C@DNA**) as a result of a constructive approach. Replication of DNA in GVs was conducted by PCR. As is well known, the temperature of the GV dispersion is raised to nearly 95 °C and is lowered repeatedly to unwind DNA duplexes (the template DNA in this experiment was 1164 bp). At first glance, such manipulations seem artificial, but terrestrial geysers, where such temperatures are readily achieved, have recently become a promising potential birthplace for life on Earth [34,35]. Given such an environment, PCR manipulation can be considered as a follow-up experiment for the origins of life.

Surprisingly, when the raw material of GV membrane **V*** was added, clear GV divisions occurred multiple times within 5 min, as observed by real-time microscopic measurement (Figure 9) [36]. Moreover, GV division is likely to occur only when DNA is amplified. In other words, if PCR was performed without template DNA, division after the addition of **V*** was slow, and the frequency was less than 10%.

## 5. Recursive Proliferative Model Cell Carrying a Cell Cycle

### 5.1. Fusion between Depleted Model Protocell with a Conveyer GV

As daughter GVs are depleted of deoxynucleotides (dNTPs) and the catalytic molecule **C** after several divisions, the depleted components must be replenished [37]. If a positively charged daughter GV and a negatively charged conveyer GV containing the depleted components are mixed, they can adhere and fuse [38,39]. However, these adhesion and fusion dynamics are not triggered by external stimuli. Therefore, it would be valuable to design an environmentally sensitive GV for vesicular fusion [40].

The membrane composition of a protocell GV is heavily inclined toward zwitterionic lipids, for example, POPC:1-palmitoyl-2-oleoyl-*sn*-glycero-3-phosphoglycerol (POPG) = 9:1, whereas that of the conveyer GV is solely composed of POPG. When the pH of the outer aqueous phase is acidic (pH 2–3), the membrane charge of a GV becomes positive because of the partial protonation of POPC, whereas that of the conveyer GV remains negative because of the remaining anionic POPG, as shown in Figure 10 [37]. Thus, lowering the pH results in membrane adhesion and fusion between the two types of GVs. The substrates, including the DNA polymerase, are then transported to the depleted daughter GV. This principle was applied to the “ingestion phase” of the cell cycle model (Figure 11). This fusion is useful for restoring the membrane charge, which shifts to the positive side, owing to the increase in cationic membrane lipid **V** by the addition of membrane precursor **V***.

### 5.2. Four Phases in a Recursive Proliferative Model Protocell

If a model protocell experiences thermal cycling as if it is beside a geyser in a hot spring, DNA amplification occurs in the protocell containing template DNA and dNTPs (replication phase shown in Figure 11). The amplified DNA in the GV is surrounded by cationic lipid **V** and dissolves in the GV membrane to form a complex (**C@DNA**) with the cationic catalyst **C**. This DNA–catalyst complex acts as a supramolecular catalyst in the GV membrane (maturation phase) to cause efficient local conversion of membrane precursor **V*** to membrane molecule **V**, and the deformed GV in the budded form undergoes GV division (division phase). This model protocell has a primitive cell cycle consisting of four phases (ingestion, replication, maturation, and division) and is capable of proliferating grand-daughter model protocells over generations (Figure 11) [37].

This primitive cell cycle is driven by external and internal stimuli, such as (1) promotion of DNA replication by thermal cycling, (2) maturation of catalytic activity by penetration of amplified DNA into the membrane, (3) promotion of cell division by high concentrations of **V***, and (4) fusion of the conveyer GV to uptake the depleted substrates by pH drop.

In contrast to Stoddart’s ring-shaped molecular shuttle [41], which moves randomly, owing to thermal energy, this model protocell is a supramolecular ratchet and demonstrates its ability to rotate in one direction through the model cell cycle because this cell cycle involves irreversible processes, such as the ingestion stage, including diffusion of substrates inside the GV. The transport of the contents from the conveyer GV to the target GV by GV fusion is equivalent to endocytosis in living cells and is crucial in the model protocell world.

### 5.3. Release Materials from a GV

Active research has been conducted on the transformation of GV morphology with UV light. One example is the UV irradiation of a GV consisting of azobenzene-incorporated lipids. The *trans*- to *cis*-photoisomerization of the azobenzene unit causes a variety of GV morphological changes, such as tubing, pearling, budding, fission [42,43], and fusion [44], which are derived from the large conformational changes of azobenzene-incorporated lipids. Light-induced reduction in the permeability of GV membranes is another interesting application. It contains a lipid-bearing nitrobenzyl moiety. Since photodegradation of *o*-nitrobenzyl-attached lipids converts it into a water-soluble amphiphile, the permeability and morphology of the GV can be photochemically regulated [45,46,47].

In contrast to the description on the uptake of substrates from the aqueous phase by GVs (Section 5.2), it is worthwhile to mention the release of contents from GV. For this purpose, a GV with a POPC/photolabile lipid to cholesterol ratio of 4:1:1 as the membrane component was prepared by Machida et al. [46]. The photolabile phospholipid bears *o*-nitrobenzyl groups at both terminals of its hydrophobic legs [48]. UV irradiation converts the terminal groups to carboxyl groups (Figure 12a). This water-soluble phospholipid escapes the GV membrane and creates pores. Subsequently, a water-soluble fluorescent dye (pyranine) was released into the outer aqueous phase through the pores, as confirmed by fluorescence microscopy (Figure 12b) [46]. The emptied GV maintained its spherical shape by self-healing, owing to the strong membrane tension of the GV membrane. This content release is essential for the removal of waste or toxic intracellular materials and can also be used for light-operated dosing as a “supramolecular delivery machine”.

If a GV is placed inside another GV, the former of which contains photo-responsive caged phospholipids in its GV membrane, a substrate in the inner GV can be released into the water phase of the outer GV when irradiated by UV light. As a model experiment, a fluorescent probe (SYBR-Green) in the inner GV was released into the water pool of the outer GV containing DNA (Figure 12c). Observation of intense green fluorescence emitted from the intercalated complex of DNA with SYBR-Green confirmed the success of the expected procedure [46]. Using this technique, the following reactions can be conducted: if two interfering reactions, oxidation and reduction, are conducted in the inner and outer GVs separately, after the termination of the reaction, the two products may be mixed in a water pool of the outer GV by UV irradiation, as schematically shown in Figure 12c.

## 6. GV Growth and Division with an Enzymatic Reaction

It has been experimentally demonstrated that enzymatic reactions can induce GV growth and division. For example, liposome growth, triggered by phospholipid production via acyltransferases inside liposomes, has been reported [49,50]. However, liposome growth based on enzymatic reactions is limited by the low production of phospholipids. The inner urease reaction could increase the inner pH and induce GV division through the decomposition of urea into ammonia because oleic acids transform into oleates and produce oleates that diffuse to form vesicles in the original GV [51,52,53]. Herein, we focus on simultaneous GV growth and division, which is promoted by the internal PCR product, amplified DNA, and works as a supramolecular catalyst with lipids for the self-reproduction of GVs.

In self-reproducible GV coupling with DNA amplification, as mentioned in Section 4.2 [36], it is hypothesized that when cationic membrane lipid **V** is mixed in the vesicular membrane, anionic DNA is surrounded by **V** (Figure 13a,b), and the surface of the DNA becomes hydrophobic. DNA is then incorporated into the GV membrane (Figure 13c), forming a complex, **C@DNA**, with the protonic catalyst **C** serving as a supramolecular catalyst (Figure 13d). The replacement of **V** with **C** proceeds because the imidazolium cation is more exposed than the quaternary ammonium cation. Local formation of **V** causes GV deformation in the budded form (Figure 13e). It has been suggested that DNA, which is essentially an information molecule, can function as a co-catalyst under such circumstances. This function would be a reward for the constructive approach.

The formation of membrane lipid **V** in the GV membrane assisted by **C@DNA** slightly resembles the activity of membrane proteins, although phospholipid biosynthesis in living cells proceeds in the membrane of the endoplasmic reticulum, which is an inner organelle [54]. The main purpose of this study was to elucidate the structure of **C@DNA** and analyze the colocalization between DNA and catalyst **C**.

### 6.1. Structural Investigation of C@DNA

To reveal the close proximity of the DNA and protic catalyst **C**, Förster resonance energy transfer (FRET) measurements were performed. This method is based on excitation energy transfer between two fluorophores in proximity. When the absorption band of the fluorescent probe of catalyst **C** (energy donor) was selectively UV-irradiated, strong red fluorescence emitted from Texas Red-DNA (energy acceptor) in the GV membrane was observed (Figure 14a) [55]. This result strongly suggested that **C**-BODIPY is adjacent to Texas Red DNA in the GV membrane.

A remarkable strand length dependence was observed for three types of DNA with different strand lengths (20, 374, and 1164 bp). As the strand length of the encapsulated DNA increased, FRET intensity increased (Figure 14b). This experimental result was unequivocally interpreted by the fact that catalyst **C** is localized around DNA, forming a DNA–**C** complex (**C@DNA**), implying multi-step energy transfer along the DNA strand [55].

To investigate the catalytic activity of **C@DNA**, the hydrolysis rates of **V*** were determined by UV-visible absorption spectroscopy in a buffered solution. The kinetic results showed that the coexistence of **C** and DNA enhanced the hydrolysis rate by approximately 10 times compared to that in the presence of either **C** or DNA. This indicates that DNA acts as a co-catalyst for **C@DNA** [55]. However, this high reactivity is lost as the reaction proceeds. This tendency suggests that the catalyst **C** surrounding the DNA was gradually replaced by an increased **V** to weaken the catalytic activity of **C@DNA**.

The dispersion of GVs with and without cationic membrane lipid **V** was performed. The GV membranes of both types of GVs were stained with BODIPY-HPC (phospholipid), and 1164 bp DNA was tagged with Texas Red. Observations using laser scanning confocal microscopy (LSCM) with high sensitivity were conducted on two types of PCR-subjected GVs [56]. Only GVs containing cationic **V** showed red fluorescent dots emitted from Texas-Red-tagged DNA in the green fluorescent GV membrane containing BODIPY-HPC (Figure 15). This provides direct evidence that DNA can enter the proliferative GV membrane containing cationic **V**.

In the colocalization analysis [57], the positive correlation between BODIPY-**C** and Texas Red DNA (dotted square in Figure 14a) was one order of magnitude higher than that between BODIPY-HPC (phospholipid) and Texas Red DNA. This is reasonable because the polar head of protic catalyst **C** is an imidazolium salt that exists as a more exposed cation than the quaternary ammonium cation. This situation leads to the spontaneous formation of **C@DNA** in the GV membrane, which serves as a lipo-deoxyribozyme, catalyzing the production of membrane lipid **V** from its precursor **V***. This experiment is a good example of the applicability of this technique for the analysis of supramolecular functional assemblies.

### 6.2. Dependence of Encapsulated DNA Strand Length on Proliferation

The experimental results of the length-dependent FRET intensity on DNA strongly suggest that the proliferation activity of GV-based model cells can be controlled by the strand length of the incorporated DNA. Here, we discuss the possibility of using DNA with three different strand lengths that exhibit different proliferation abilities.

For this experiment, an anionic PG tagged with polyethylene glycol (PEG) (DSPE-PEG 1000; 22-mer ethylene glycol) was used to emphasize the effect of different DNA strand lengths on the self-reproduction of GVs [55]. Incidentally, PEG is not only a steric hindrance to the encapsulated DNA but also a poor solvent that excludes DNA (Figure 16). The resulting PEG-loaded GV membranes were labeled with different fluorescent probes (0.1%) to distinguish between the different strands of DNA encapsulated in the GVs.

Three types of GVs were prepared for confocal microscopy measurements: GVs containing three different DNA strand lengths (374 bp (S), 1164 bp (M), or 3200 bp (L)) with approximately equal amounts of nucleotides. The increased ratio of the number of PCR-subjected GVs after the addition of **V*** was counted by confocal microscopy. GVs with a diameter of less than 5 μm were excluded as protocells because of insufficient volume for enzymatic reactions. As shown in Figure 17a, the time dependence of the increased ratios was significantly dependent on the strand length of encapsulated DNA. GV (S) grows fast but decreases later by further splitting; GV (M) grows fast, and the increase is saturated at 1 h; and GV (L) grows slowly but catches up with GV (M) after 1 h [55]. The different time dependences of the increased ratios of the three types of GVs are shown in Figure 17b.

In addition to confocal microscopy measurements, flow cytometry measurements of GVs with cell numbers of 10^3^–10^4^, and microscopic tracings of the morphological changes of single GVs containing DNA with different strand lengths were conducted on 30–40 specimens. There were no inconsistencies between the three independent experiments. These results clearly indicate that the proliferative ability of GV-based model cells is strongly influenced by the strand length of the encapsulated DNA.

## 7. Emergence of Primitive Information Flow in a Model Protocell

### 7.1. Role of Lipo-Deoxyribozyme in Primitive Information Flow

Considering the catalytic activity of **C@DNA** in the hydrolysis of **V*** to **V** in the GV membrane, it should be noted that the function of **C@DNA** is analogous to that of RNA, which forms a complex with metal ions to promote the hydrolysis of RNA and other substrates. This RNA function is referred to as ribozyme (Figure 18a). C@DNA efficiently catalyzes the hydrolysis of the precursor of the membrane lipid (**V***), which is an imine-type bolaamphiphile, especially when the substrate is sandwiched between imidazolium catalysts, as proposed by Breslow’s mechanism (Figure 18b) [58]. Therefore, it is rational to call **C@DNA** in a GV membrane a lipo-deoxyribozyme [55,59].

Although these results strongly suggest that DNA strand length influences the activity of lipo-deoxyribozyme, elaborating whether DNA sequence affects proliferation is interesting from the perspective of the emergence of biological information. Hence, reference experiments were conducted using DNAs with different base sequences of similar length, and it was found that the proliferation ratios for all DNAs examined were almost the same. This supports the interpretation that DNA strand length affects the rate of GV increase [55,59].

The conventional “central dogma”, in which information coded in a base sequence of DNA is transmitted to RNA and protein via transcription and translation, respectively, strictly regulates the flow of information and the activation of metabolism. However, it is difficult to believe that such advanced information transmission, consisting of a transcription–translation system that modern living cells possess, has been in place since the beginning of life. Alternatively, primitive polymers may have regulated the proliferation of protocells by means of a more general physical cause, such as strand length, during a certain stage of biological history. In this case, the primitive information flow is significantly shorter than that of the conventional central dogma, as shown in Figure 19 [59].

### 7.2. Phenotypic Plasticity of Model Protocells

Different organisms have different genotypes and phenotypes. A genotype is a combination of genes that gives rise to a trait, such as AA or Aa, and determines the morphology of an organism. Hence, an organism may have different phenotypes even if the genotypes are the same. Phenotypic diversity is also known as phenotypic plasticity. Species that fit the environment are often the dominant species. A straightforward example is as follows: a certain aquatic microorganism is cylindrical in shape in a nutrient-rich environment but becomes multi-tubular to increase its surface area when nutrients are scarce [60]. In primitive cells, where the flow of information is not as precise and robust as in present-day organisms, phenotypic diversity would have occurred more frequently and influenced evolution.

Recently, a model of phenotypic plasticity in the prebiotic era was proposed. In this case, the starvation time is an environmental factor associated with phenotypic plasticity [59]. In addition to the budding-type deformation, which is a typical deformation pattern (phenotype) of the proliferative dynamics of the current model protocell, a new phenotype, multi-tubulation-type deformation, appeared when the membrane precursor **V*** was added after a long starvation time (Figure 20a). The appearance of the multi-tubular phenotypes is restricted by the ratio of POPG in the membrane. Notably, it was in the range of 24–39 mol%, suggesting that the appearance of the multi-tubular phenotypes require a relatively strong electrostatic interaction between the negative surface charge of the GV and the cationic nourishing lipid precursor **V*** [59]. This phenotypic diversity may be due to the distribution of **C@DNA** in the GV membrane. If **C@DNA** is formed in an isolated manner, newly formed membrane lipid **V** can diffuse in all directions. In such situations, it is difficult for budding deformation to occur. If membrane areas surrounded by two **C@DNA**s appear within a relatively short starvation time, budding deformation may be induced. However, when the starvation time was long, many such areas were found in the GV membrane, inducing multi-tubulation of GVs (Figure 20b) [59].

### 7.3. Expression of Predominant Species in Competitive Proliferation

As outlined in Section 6.2, it was clear that the increased ratios of the three types of model protocells were distinctly different depending on the encapsulated DNA strand length [55]. However, in nature, various species compete with each other, and the dominant species emerge after “natural selection”. Therefore, it is worth estimating the increased number of coexisting types of GVs in the same container when **V*** is added (Figure 21) [59].

As shown in Table 1, the ratio of the rate of increase in GV (M) and GV (L) in separate containers after one hour was 1.1 (Table 1), whereas the ratio increased to 1.8 in the case of competitive proliferation (Table 2). In the case of GV (M) and GV (S), however, the ratio of increase after one hour was 2.4 under noncompetitive conditions (Table 1), whereas the ratio of competitive proliferation was only 1.1 (Table 2). This was presumably because the two types of GVs (GV (M) and GV (S)) coexisting in the container compete for **V***; therefore, proliferation stops before sufficient growth has been achieved. When a sufficient amount of **V*** was added, the dominance of GV (M) increased 1.8-fold, approaching that in the noncompetitive case (Table 3).

This competition appears to occur in the biological world. If various GVs containing DNA of different strand lengths are mixed in the same container, competition will emerge among them, and eventually the GV with the most hostile proliferation characteristics will survive as the dominant species. This means that a “primitive natural selection” was created in a laboratory flask (Figure 21) [59].

### 7.4. Environment-Sensitive and Evolvable Oil Droplets

Autonomous movement is one of the most characteristic life properties. It has been reported that an oil droplet of oleic acid anhydride self-propelled in alkaline water, driven by hydrolyzed oleic acid, works as a surfactant to decrease the surface tension of the droplet, inducing Marangoni convection [61]. As an extension of this line, sustainable self-propulsion of oil droplets [62], chemotactic [61,62], and phototactic oil droplets [63] have been reported. Thus, oil droplets can be regarded as naïve model protocells.

Taylor et al. presented an oil-in-water droplet system comprising an amphiphilic imine dissolved in chloroform that catalyzes its own formation by combining hydrophilic and hydrophobic precursors, which leads to repeated droplet division [64]. Reactants **A** and **B** with complementary recognition sites react together to form template **T**. **T** selectively binds **A** and **B** by the corresponding recognition sites, providing a ternary complex **ABT**. Reactants **A** and **B** are weakly amphiphilic, but by combining hydrophobic and hydrophilic groups, amphiphilic **T** is formed. Amphiphile **T** stabilizes the reverse micelles of water in chloroform. This stabilization of the droplet chloroform/water interface by **T** allows droplets to undergo fission. As the authors pointed out in their paper, if they use a bilayer vesicle as a reaction medium, the reaction could occur in an aqueous solution, resulting in linked template self-replication and compartmentalization.

Cronin et al. developed a new collaboration between self-reproductive oil droplets and liquid-handling robots. The robot creates droplets by mixing four different compounds (1-octanol, diethyl phthalate, 1-pentanol, and either octanoic acid or dodecane) in different ratios and placing them in a Petri dish [65]. The behavior of the droplets (rapid movement, binary fission, and vibration) was recorded using a camera. Analysis and theoretical modeling of the data afforded fitness landscapes analogous to the genotype–phenotype correlations found in biological evolution.

Parrilla-Gutierrez and Cronin et al. further developed a fully automated fluid-system-type robot and constructed an experimental system that could arbitrarily change the environment for evolutionary experiments, in which oil droplets are driven, deformed, and split [66]. The robot analyzed the composition of the droplets and repeated the experiment by modifying it. The results showed that selection pressure was allotted to the environment when the oil droplet composition was assumed to be the genotype. Cronin’s oil droplets certainly evolve by raising the fitness landscape, but the genotype completely depends on the composition of the oil droplets. Since the genotype and phenotype thoroughly respond to each other, freedom of phenotypes is limited, and freedom of the phenotype is lacking compared with those of life. If they enable the inclusion of a polymer with high phenotypic freedom, proliferation may produce phenotypic plasticity similar to the evolution process of life.

## 8. Proliferating Coacervate Droplets

### 8.1. Spontaneous Formation of Aggregated Polymers Obtained from Amino Acid Thioesters

Oparin and Haldene proposed a chemical evolution scenario in which polymers, generated from small organic molecules in a prebiotic environment, form proliferative self-aggregates, thus becoming primitive cells [67,68]. However, the formation of polymers and their aggregations have been studied under different conditions, and a primitive model system in which aggregates of polymers proliferate by the addition of nutrients has not been realized in roughly 100 years. Matsuo and Kurihara demonstrated a proliferating coacervate droplet via autocatalytic self-reproduction at room temperature and atmospheric pressure, resulting in experimental substantiation of the chemical evolution scenario [69].

In this study, thioesterified cysteine was designed and synthesized as a precursor of the prebiotic monomer. When the precursor was added to water, in which the reducing agent was dissolved, peptides (up to at least tetramers) were spontaneously synthesized, and a hydrophobic compound (benzyl mercaptan; BnSH) was released as a by-product (Figure 22a). This oligomerization autocatalytically proceeded at the interface between the water and droplet, the latter of which was composed of the generated peptide and benzyl mercaptan (Figure 22b).

### 8.2. Emergence of Sustainable Proliferating and Self-Maintaining Droplet Protocells

Accompanied by the oligomerization of amino acid thioesters, droplets that act as catalysts for the production of their own building blocks are formed by the liquid–liquid phase separation. While the intermittent addition of amino acid thioesters as nutritional and physical stimuli (shear stress) to the droplets allowed the droplets to autocatalytically self-reproduce and divide, the droplet-type protocell proliferated sustainably, generating distant descendants (Figure 23). When RNA is dissolved in a droplet dispersion, the droplet acquires resistance to the dissolution of lipids by the coexistence of RNA and lipids via heterogeneous condensation. RNA and lipids are condensed around the interface and center of the droplet, respectively [69]. The selectively high permeability of a proliferating peptide-based droplet could bridge the gap between the peptide, lipid, and RNA worlds in the origins of life.

The membraneless structure of droplet-based protocells facilitates selective condensation and release with high permeability, which in turn facilitates processes such as localization, colocalization, specific activation/inactivation, hypercatalysis, and sequential reactions. All these processes were spontaneously realized in an experimental system of proliferating coacervate droplets [69]. Stabilization via selective condensation induced by a membraneless structure may be essential for hydrolysis-susceptible RNAs in the protocell to allow RNA to express their functions.

Such high substrate permeability of membraneless structures is a double-edged sword when considering the evolution of protocells. High permeability facilitates the exchange of the substrate between the protocells. This exchange of substrates permits the presence of parasites in the RNA/DNA self-replicating system and inhibits the elongation and sophistication of RNA/DNA. In the early stages of life, the membraneless structure, such as the interface of a droplet, might have played an important role in the emergence of cooperation between the self-reproduction of peptide-based droplets and the self-replication of RNA/DNA, leading to the emergence of genetic information. In subsequent stages, the membrane structure, such as the vesicular membrane, would have enabled the sophistication of the genetic information coded in the RNA/DNA. Therefore, membrane generation from a proliferating membraneless droplet should be reconstructed from the perspective of the origin of life. Consequently, this autocatalytically self-reproducing droplet formulates the “droplet world” hypothesis in the origins of life, which facilitates the emergence of the protocell and the development of its complexity with the physicochemical properties of the droplet. This leads to vesicle-based protocells via coupling with information molecules (RNA and DNA), lipids, and peptide worlds in the proliferating droplet [69].

## 9. Outlook towards Evolvable Model Protocell

As described in Section 6, the growth or division of a compartment is often induced by enzymatic reactions. In the vesicle-based model protocell described above, non-enzymatic self-reproduction of the compartment, linking with the enzymatic amplification of DNA, has been successfully achieved. GV growth and division were induced by a supramolecular catalyst, called lipo-deoxyribozyme, which was formed spontaneously in the GV membrane. The lipo-deoxyribozyme acts effectively as a supramolecular catalyst for hydrolysis of membrane molecule precursors, causing local deformation and inducing GV division. This function is rather similar to that of the coenzyme proposed by Sharov, described in Section 4.1.

However, in such a GV-based model protocell, there is still room for improvement in the amplification of information molecules. Since self-replication of information is a characteristic property of life, the propagation of information molecules achieved by a non-enzymatic self-replication mechanism, originated by Orgel and Kiedrowski, is significant for protocell research, as in the non-enzymatic self-reproduction of the compartment [70]. As described below, elaborate self-replicating systems have been developed.

Kiedrowski proposed a simple model of self-replication. **P** and **Q** are monomeric molecules for self-replication [71]. Because a template molecule **T** (**P**-**Q**) is self-complementary, it can incorporate its components **P** and **Q** in a complementary manner by intermolecular forces and generate a three-molecular complex **M**. In this complex, since **M**, **P**, and **Q** are fixed in a position where they can easily react, **P**-**Q** is generated. Hence, the number of **P**-**Q** doubles as **M** decomposes. By repeating this process, the molecule **P**-**Q** can be amplified.

After a successful demonstration of an autocatalytic chemical system [71], Kiedrowski and Sievers carried out a pioneering self-replication experiment using 5′-terminally protected trideoxynucleotide 3′-phosphate (Me-CCG-p) (p means a protecting group) and a complementary 3′-protected trideoxynucleotide (CGG-p’). These two self-complementary trideoxynucleotides were reacted in the presence of the water-soluble carbodiimide EDC (l-ethyl-3-(3-dimethylaminopropyl)-carbodiimide) to yield the self-complementary hexadeoxynucleotide (Me-CCG-CGG-p’) (**H**) as a template. Using template **H**, they succeeded in synthesizing a complementary hexamer [72].

Non-enzymatic amplifications of information molecules have frequently been carried out in the prebiotic approaches. Szostak’s group reported self-replicated short DNA (≈30 bp) in a prebiotic cell, as mentioned in Section 4.1 [33]. Recent reports of non-enzymatic template-directed replications on specific RNA or peptides also have the potential of being involved in evolvable supramolecular machines [73,74,75,76,77].

Enzymatic self-reproduction of the compartment and enzymatic replication of the information molecule are regarded as the two pillars of biological evolution. Stano has already pointed out the information storage and freedom of change in model protocells [3], and these features are certainly related to their evolvability. Enzymatic replication systems may have less room on the freedom of change than non-enzymatic ones because the enzymes are already results of the biological evolution. Therefore, the coupling of non-enzymatic self-reproduction of the compartment with non-enzymatic self-replication of the information molecule in a manner of dynamic supramolecular chemistry breaks a new field in the development of evolvable materials, e.g., the ultimate supramolecular machines without limitation of biological evolution. This issue remains an open question that can be tackled by the constructive approach using supramolecular machines.

## 10. Conclusions

The intrinsic property of life emerges from cooperative dynamics performed not by individual biomolecules but by interacting biomolecules (and their assembly) [1,2,3]. When the horizontal axis represents the complexity of a GV-based model protocell, and the vertical axis represents the pattern of membrane dynamics, the hierarchy of membrane dynamics of the model protocell increases from the self-production of a membranous compartment (e.g., a GV) [9,10] to a linked proliferation between reproduction of a GV and replication of information molecules, such as RNA and DNA [30,31,36], and then further to a recursive proliferation of a model protocell [37] (Figure 24).

A spontaneously formed supramolecular catalyst, which is a DNA complex with the protonic catalyst **C** (**C@DNA**), serves as a lipo-deoxyribozyme and accelerates the formation of membrane lipids in the GV membrane. In addition, the division frequency of a model protocell depends on the DNA strand length in **C@DNA**. This “cause and effect” may be regarded as the “primitive flow of information”. As a more direct third regulation by DNA, DNA-strand-length-dependent regulation is also present. For example, the catalytic activity of a lipo-deoxyribozyme is dependent on its DNA chain length. A recent phenomenon found in living cells is the DNA-chain-length-dependent formation of a droplet via liquid–liquid phase separation. This phase separation regulates the activity of the immune system [78,79]. The discovery of new life phenomena or the development of antiviral drugs may be realized by focusing on DNA-chain-length-dependent formation and the activity of a supramolecular catalyst constructed with sequence-free DNA.

This phenomenon is crucial for an autonomous model protocell that responds to inner and outer environments. This information flow induces phenotypic plasticity with respect to starvation time as an external stimulus [59]. Moreover, the appearance of a predominant species from the competitive proliferation of model protocells containing DNA with different strand lengths is expected to lead to an evolvable model protocell [59]. As evolution is the upper level of hierarchical dynamics in proliferation, it enables the production of adaptive species in a given environment (Figure 24).

Recently, Matsuo and Kurihara reported a proliferating-peptide-based droplet created using synthesized amino acid thioesters as the prebiotic monomers. This novel membraneless peptide-based droplet formalizes the “droplet world” in the origins of life and contributes to a new trend in the development of membraneless model protocells from the aspect of autocatalytic self-reproduction on the basis of the characteristic features of the membraneless structure, as discussed in Section 8.2 [69].

As mentioned in the introduction, a cell that is the smallest unit of life can be considered as the ultimate “supramolecular machine”. Research on artificial cells has made unexpected progress in GV-based artificial model protocells, such as self-reproduction of a compartment, proliferation of GV-based protocells containing DNA, primitive signal transduction in protocells, environment-responsive protocells, and primitive natural selection among different species. These model cells can be regarded as supramolecular structures. Compared with conventional molecular machines, supramolecular machines have a hierarchical structure, and autonomous moving machines can also be constructed. These results contribute not only to the development of the artificial protocell world but also to the field of micromachines and micro-robotics, which have been rapidly advancing in recent years. The concept of adaptation to a given environment and evolution that emerged from natural selection may open a new horizon for intelligent supramolecular machines and robotics [59,64,65,66,70].

## Figures and Tables

**Figure 1 life-12-01635-f001:**
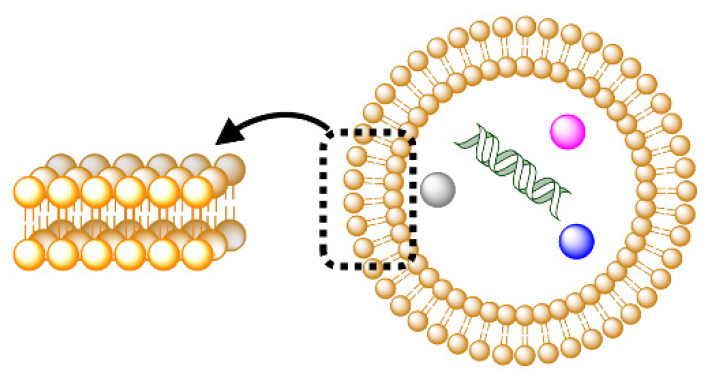
Spontaneous formation of giant vesicle encapsulating ions and hydrophilic molecules.

**Figure 3 life-12-01635-f003:**
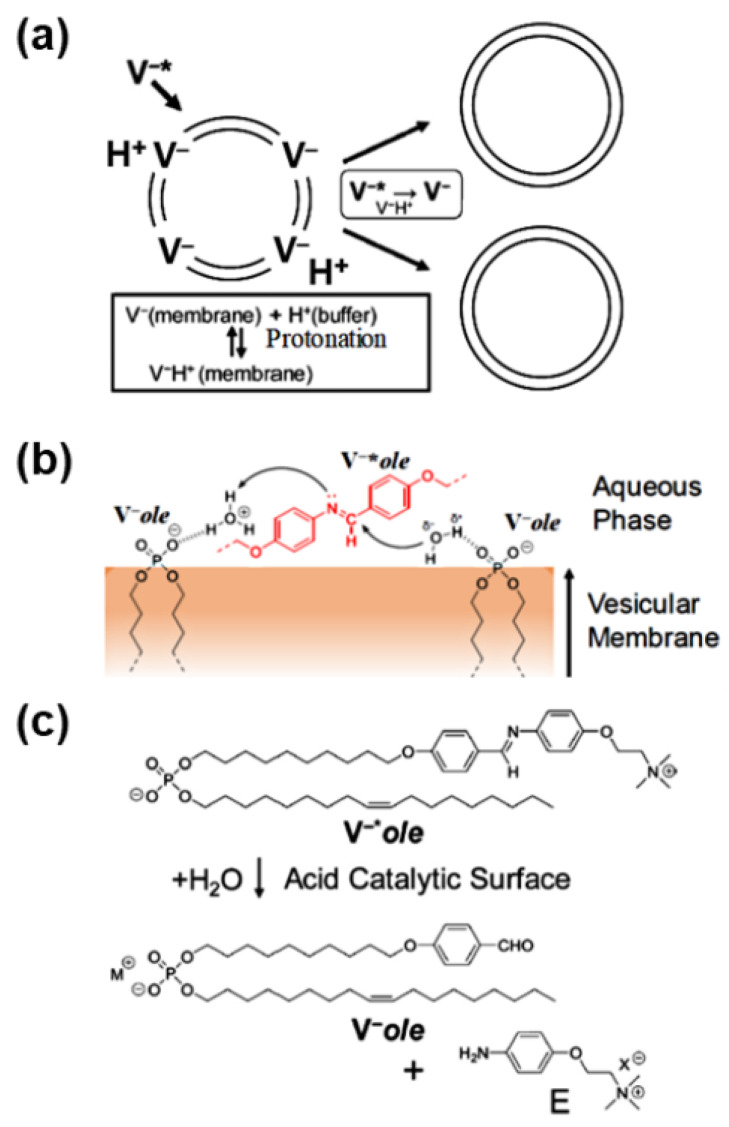
(**a**) Sustainable proliferative phospholipid GV-based model protocell. (**b**) Schematic illustration of catalytic hydrolysis of **V^−^*_ole_** operated by an oxonium ion (H_3_O^+^) and a water molecule, supported by anionic phosphate molecules. (**c**) Molecular structure and hydrolysis scheme of **V^−^*_ole_** and **V^−^_ole_**.

**Figure 4 life-12-01635-f004:**
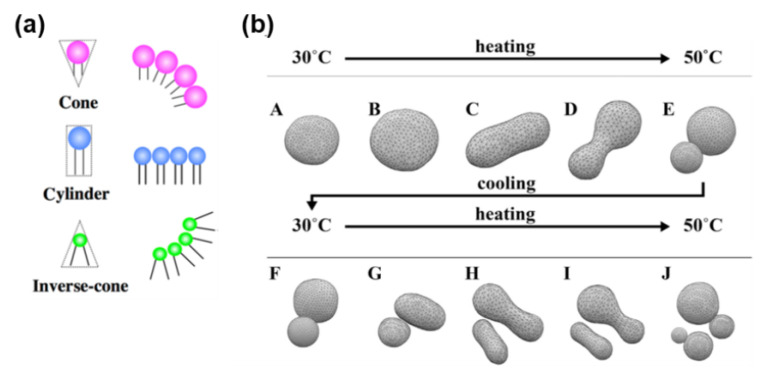
(**a**) Amphiphiles with increasing *p* values. The figure was adopted from [23]. (**b**) ADE analysis of the morphological change from spherical form to limited form by heating. The amphiphile splits at the re-heating stage. Adapted with permission from Ref. [24]. Copyright 2016 Elsevier.

**Figure 5 life-12-01635-f005:**
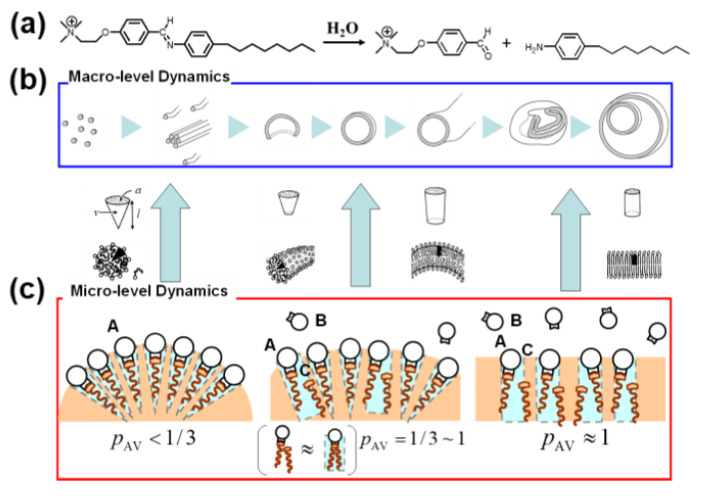
Time-dependent morphological change from micelles to giant vesicles partially composed of loose complexes between the original amphiphile and the hydrophobic product. (**a**) Reaction scheme of hydrolysis of *N*-benzylidene aniline derivative. (**b**) Gradual morphological changes of hybrid molecular assemblies (from micelle, through tubular micelle and vesicle, to nested giant vesicles) containing a hybrid complex; the ratio of the original amphiphile decreased, while that of the hydrophobic product increased. (**c**) Temporal change of the morphologies of hybrid molecular assemblies associated with the gradual change of the packing parameters.

**Figure 6 life-12-01635-f006:**
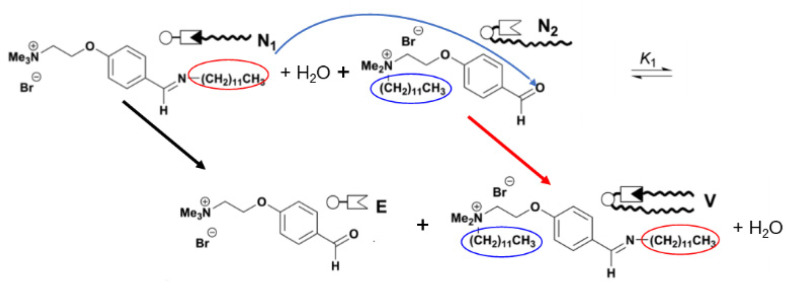
Migration of dodecylamino group from **N_1_** to **N_2_** to form a two-legged amphiphile **V** and electrophile **E**.

**Figure 7 life-12-01635-f007:**
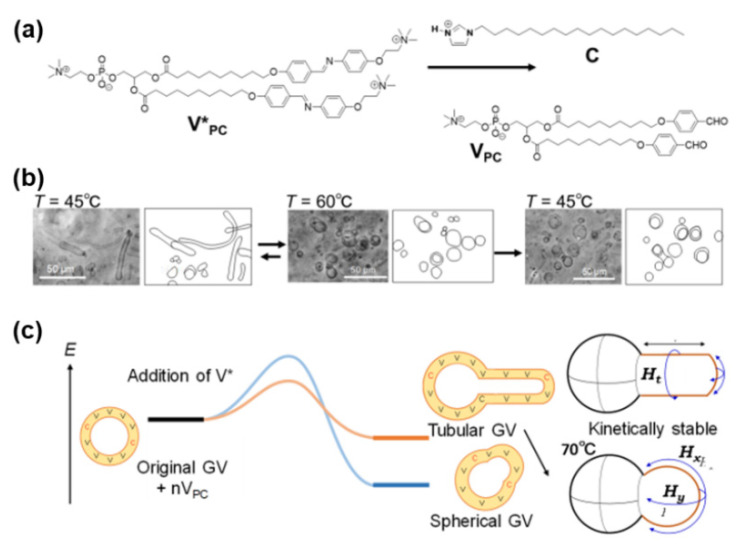
(**a**) Hydrolysis of **V_PC_*** produced membrane lipid **V_PC_** by the action of catalyst **C**. (**b**) Tubular GV_PC_ at 45 °C transformed to spherical GV_PC_ after heating at 60 °C and maintained its shape after cooling at 45 °C. (**c**) Elastic energy profile of morphological changes from tubular to spherical GV_PC_s at 60 °C. Adapted with permission from Ref. [27]. Copyright 2019 The Chemical Society of Japan.

**Figure 8 life-12-01635-f008:**
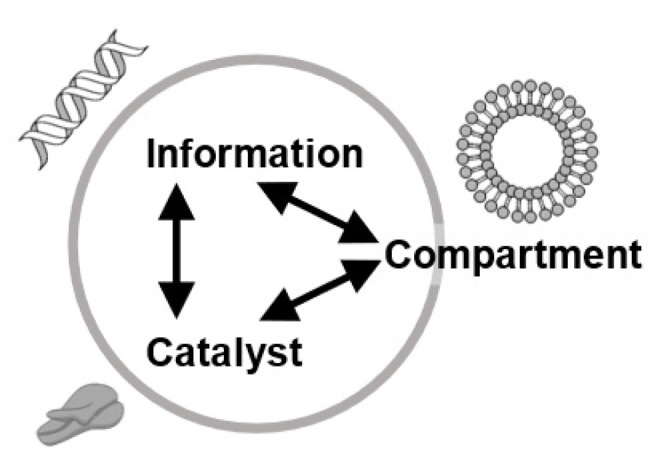
Three fundamental elements for a model protocell: compartment, information molecule, and enzyme.

**Figure 9 life-12-01635-f009:**
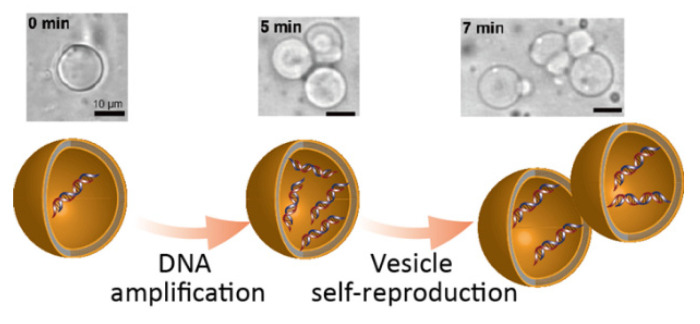
Self-proliferative model protocell. Linkage between GV self-reproduction and replication of encapsulated DNA. Adapted with permission from Ref. [36]. Copyright 2011 Springer Nature.

**Figure 10 life-12-01635-f010:**
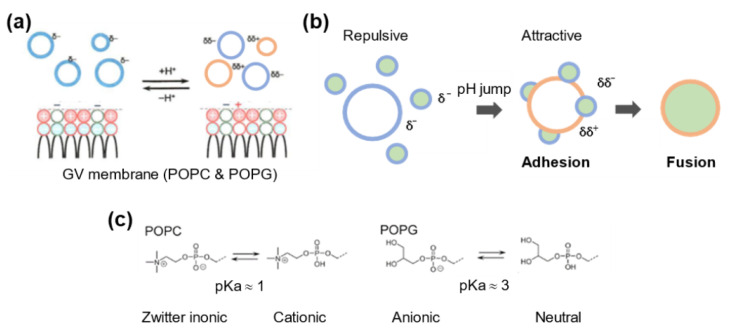
Vesicular fusion between a depleted GV and a conveyer GV. (**a**) Charges of phospholipids (POPC and POPG) and surface charges of depleted GVs and conveyer GVs in buffered dispersion (left) and those in acidic dispersion (right). (**b**) Adhesion and fusion between two typed GVs in acidic dispersion (pH = 2~3). (**c**) Charged structures of hydrophilic head groups of POPC and POPG.

**Figure 11 life-12-01635-f011:**
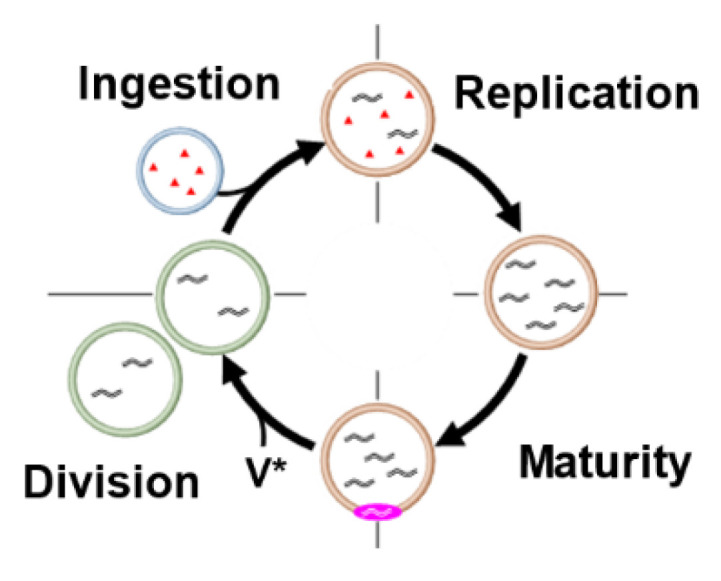
The recursive proliferation of a model protocell with four discrete phases.

**Figure 12 life-12-01635-f012:**
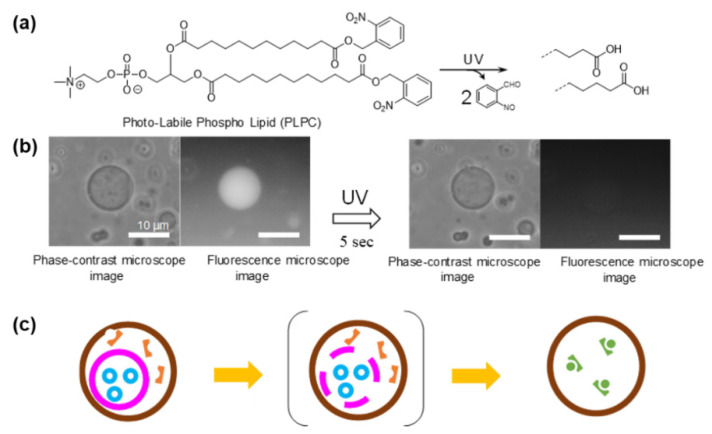
(**a**) Photodecomposition of caged phospholipid. (**b**) Phase-contrast and fluorescence microscopic images of GVs before and after UV irradiation. (**c**) Photo-irradiation of inner GV in the outer GV releasing product (inner) to the water phase of the outer GV-containing product (outer). They react to produce the final product.

**Figure 13 life-12-01635-f013:**
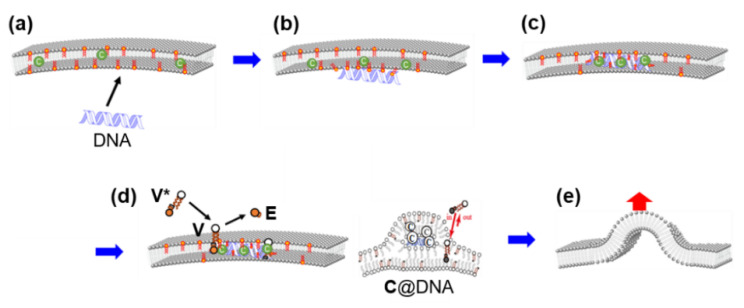
(**a**) Adhesion of DNA on a GV surface containing cationic **V**. (**b**) Wrapping DNA by cationic **V**. (**c**) Intrusion of DNA into GV membrane and replacement of **V** by **C**. (**d**) Local formation of **V** from **V*** aided by **C@DNA** (right; expanded figure). (**e**) Budding deformation caused by local membrane production at **C@DNA**.

**Figure 14 life-12-01635-f014:**
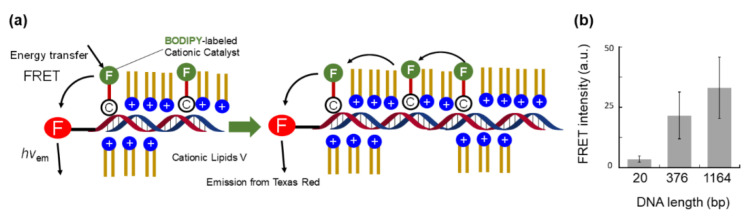
(**a**) FRET from **C**-BODIPY to Texas Red DNA, showing the proximity between DNA and catalyst **C**. (**b**) FRET intensity vs. the length of encapsulated DNA. Adapted with permission from Ref. [55]. Copyright 2019 Springer Nature.

**Figure 15 life-12-01635-f015:**
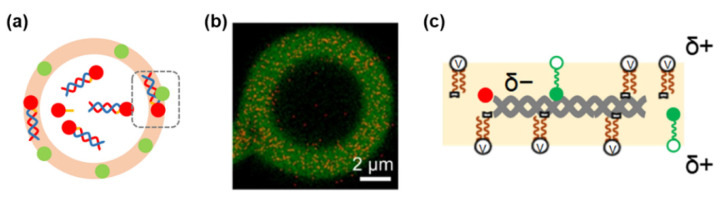
(**a**) Colocalization analysis using Texas Red (red spot)-tagged DNA, and BODIPY (green spot)-tagged catalyst. The figure was obtained from [55]. (**b**) LSCM image of PCR-conducted GV with cationic **V**. Red fluorescent spots emitted from Texas-Red-tagged DNA and green fluorescent GV membrane stained with BODIPY-tagged HPC (hexadecanoyl-*sn*-glycero-3-phosphocholine). Reprinted with permission from Ref. [56]. (**c**) Schematic illustration of **C@DNA**.

**Figure 16 life-12-01635-f016:**
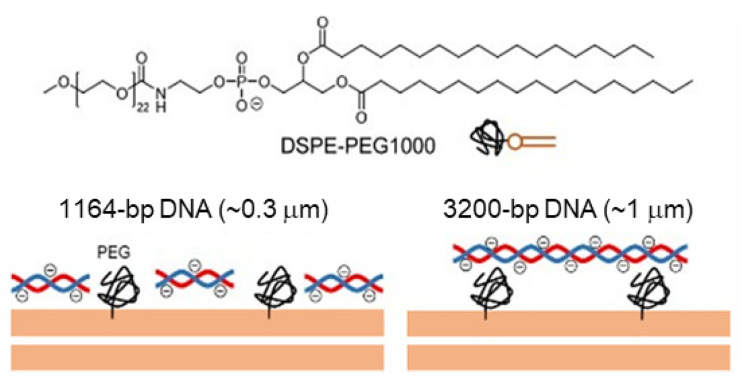
Inhibition of DNA proximity to cationic membranes by DSPE-PEG1000.

**Figure 17 life-12-01635-f017:**
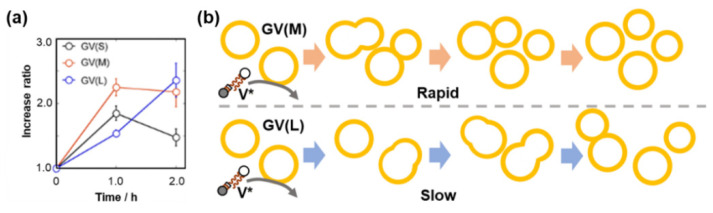
(**a**) Time dependence of increased ratios of the proliferation of GV (S), GV (M), and GV (L). Adapted with permission from Ref. [55]. Copyright 2019 Springer Nature. (**b**) Schematic drawing of the ratio of proliferation processes of GV (S), GV (M), and GV (L).

**Figure 18 life-12-01635-f018:**
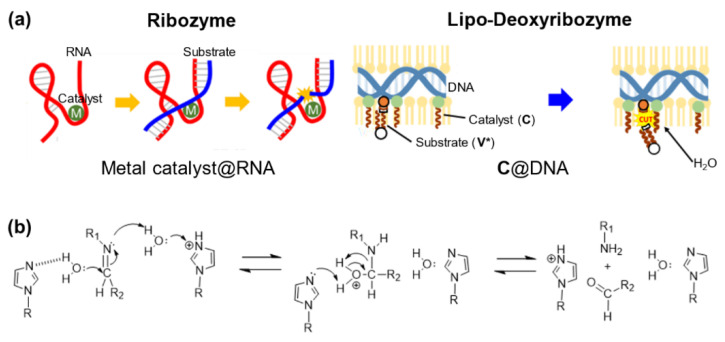
(**a**) Structure and catalytic function of ribozyme and similarity of **C@DNA** as a lipo-deoxyribozyme. (**b**) Breslow’s push and pull mechanism of hydrolysis of imine derivative. Adapted with permission from Ref. [55].

**Figure 19 life-12-01635-f019:**
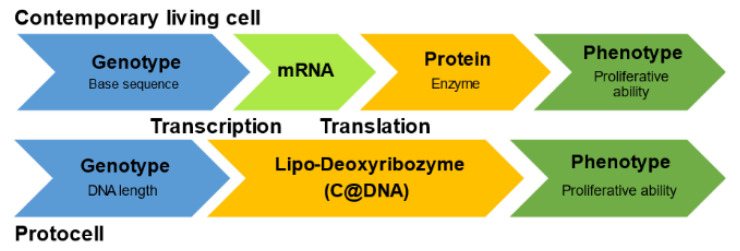
The primitive flow of information in a model protocell. No transcription–translation system is present in the current model protocell.

**Figure 20 life-12-01635-f020:**
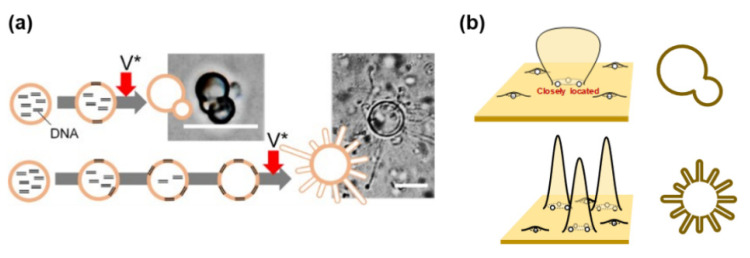
Expression of phenotype plasticity. (**a**) Budding deformation occurs after a short starvation time. Multiple-tubulation appears after a long starvation time. Adapted with permission from Ref. [59]. (**b**) If the concentration of **C@DNA** in a GV membrane surface is low, a budding deformation occurs, while a high concentration leads to a multi-tubulated GV.

**Figure 21 life-12-01635-f021:**
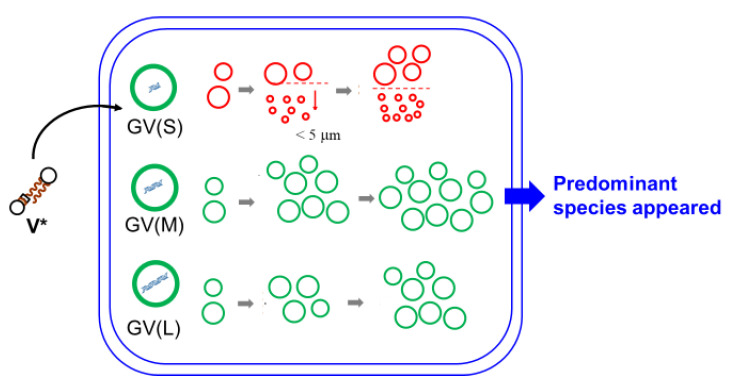
Dependence of proliferative efficiencies of GV-type model cells on the different strand lengths of the internal DNA. Competitive proliferation leads to the emergence of a predominant species.

**Figure 22 life-12-01635-f022:**
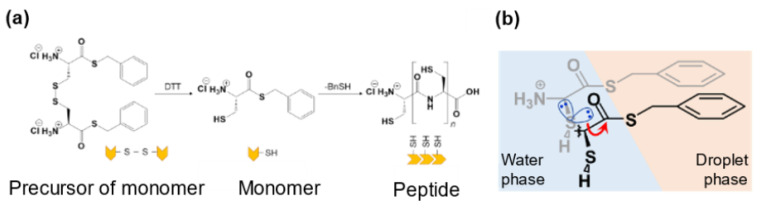
(**a**) Reduction of the precursor by dithiothreitol (DTT) produces a monomer, and monomers oligomerize to produce peptides. (**b**) Oligomerization of monomers occurs at the interphase between the water phase and the droplet phase.

**Figure 23 life-12-01635-f023:**
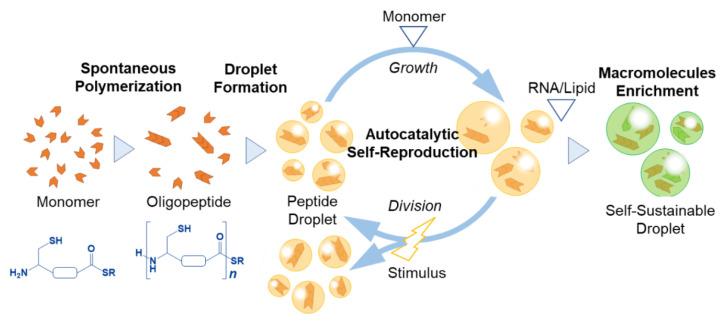
Emergence of proliferating droplet protocells. In the first stage, the amino acid thioester is oligomerized to produce a peptide. Droplets were made of the hydrophobic product generated by liquid−liquid phase separation. Continuous addition of the amino acid thioesters as a source of nutrition and physical stimulus to the droplets allowed the droplets to divide. The droplet that concentrated RNA was robust against the dissolution with the lipid.

**Figure 24 life-12-01635-f024:**
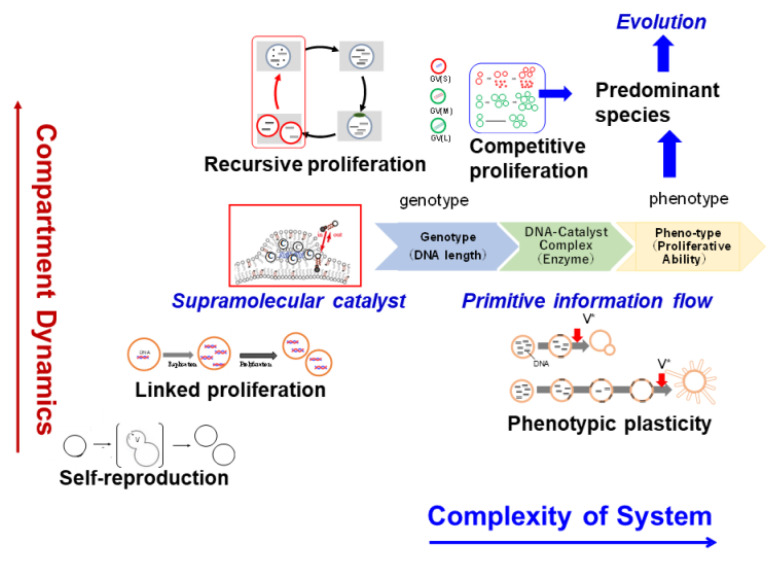
Correlation between the complexity of the system and compartment hierarchical dynamics of model protocells.

**Table 1 life-12-01635-t001:** Increase ratios of GV(L)/GV(M)/GV(S) under noncompetitive condition.

GV	Noncompetitive Increase Ratio
L	125 ± 13%
M	141 ± 14%
S	59 ± 9%

**Table 2 life-12-01635-t002:** Increase ratios of GV(M)/GV(S) under competitive condition.

GV	Competitive Increase Ratio
L	76 ± 7%
M	133 ± 7%
M	75 ± 8%
S	69 ± 5%

**Table 3 life-12-01635-t003:** Dependence of Increase ratios of GV(M)/GV(S) on V* concentration under competitive condition.

(Total Lipid)/V*	GV(M)/GV(S)
3/1	1.1 ± 0.10
1/1	1.2 ± 0.11
1/3	1.8 ± 0.20

## Data Availability

Not applicable.

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
