# Peer review of "Evolution of Proliferative Model Protocells Highly Responsive to the Environment"

_life, 2022, doi:10.3390/life12101635_

Round 1

Reviewer 1 Report

The article reviews different strategies to generate proliferating model protocells. It is well-written and comprehensive, even for non-experts in the field.

In general, I feel that the literature is not sufficiently complete and that the explanations sometimes lack precision.

The authors explain that the different strategies are discussed using representative case studies. This is a good approach. However, I believe that in a review like this, the authors should at least cite the other examples (without detailing them). This would also help the reader understand whether the synthesis strategies have been studied frequently or only once/twice.

It would be interesting if the authors could include a small outlook section on existing strategies, supported by references.

Figure reference needs a deep check in all text. For example, figure 1b in text should be figure 1c, etc etc.

Reviewer 2 Report

In this paper the authors presented a review and a summary of (mainly) their findings about the proliferative protocell model. In general, the manuscript is well written and reports a few systems potentially relevant for the origin of life studies. The importance of these systems is testified by the international resonance that previous publications of the authors gained and the topic is certainly suitable for the special issue.

I believe that the manuscript could be accepted for publication in Life after some issues will be addressed:

-       Though the manuscript is well organized and easy to follow, there are some sections that somehow, in my opinion, do not fit well to the rest of the paper. For example, in section 3 the authors briefly introduced the ADE theory, but this concept is never reprised in the rest of the text, so it is not clear how and when ADE theory is important to understand the division and replication of the protocells. I would suggest to recall this concept whenever appropriate in the description of the proliferative processes.
Also, the authors introduced the catalytic mechanism of C@DNA in section 4.2 and reprised it in more details in Section 6. Therefore, some concepts are repeated twice. I suggest the authors to join these sections to improve the reading flow;

-       The cited literature is appropriate; however, the authors should briefly mention other works when relevant. For example, when speaking about light induced cargo release, they might refer also to the work of Prof. Lohmüller group about azolipids. When speaking about enzymatic induced divisions they might report the work of Prof. Rossi group about urease induced budding. For the ADE theory the work of prof. Lipowsky and Prof. Seifert has to be cited. Etc.

-       P. 2 Lines 46-50: I do not understand how this paragraph fits in the introduction. It seems that some parts of the text are missing. Also, the sentence at line 46 does not make sense. Please, double check;

-       P. 2 line 100-102: I do not understand the meaning of this sentence. Please rephrase;

-       P. 3 line 105: The flow cytometry is shown in Figure 1c and not in Figure 1b;

-       P. 4 line 134: The symbol V-*ole is not defined, and also V-ole later in the text. Please, provide a definition whenever a new symbol or acronym is introduced;

-       P.5 line 163: “the inner and outer membranes” should be “the inner and outer leaflets of the membrane”;

-       P. 5 line 166: in the formula of the filling parameter symbols are not defined;

-       P.5 line 186: the filling parameter is now called packing parameter. Please uniform the definitions (symbols not defined again).

-       Figure 4: panels are not reported in the figure. Please, also provide a more descriptive caption;

-       P. 6 line 211 between brackets: C is not defined along the text (only in the figure);

-       P. 7 line 237: “protocell cell” should be “protocell”;

-       P. 7 line 253: “corporation” should be “cooperation”;

-       P. 8 line 285: the catalyst is referred as “C”, but at the beginning of the section it was defined as “Cs”. Are those molecules different? Please double check. Please, also double check along all the text (e.g. section 6.1);

-       I think that Figure 14 and Figure 15 can be joined to make the text more concise;

-       Section 7.3: please provide a definition for competitive proliferation and highlight the differences with the “noncompetitive” case;

-       P. 15 line 586: “Haldane” is misspelled;

-       Figure 23: please, in the caption, add a brief description of the RNA concentration part;

Reviewer 3 Report

This article discussed various methods to reproduce the life dynamics using a constructive approach.  It is very informative and helpful. The article is well contracted and written. I would like to recommend to publish as it is.
